# Prospective Observational Study of De-Escalation of Empirical Antibiotics in Fiji’s National Hospital

**DOI:** 10.3390/antibiotics14020124

**Published:** 2025-01-24

**Authors:** Tracey Young-Sharma, Shitanjni Wati, Vikash Sharma, Ravi Naidu, Deborah Tong, Adam Jenney

**Affiliations:** 1Medical Unit, Colonial War Memorial Hospital, Ministry of Health and Medical Services, Suva, Fiji; shitashilawati@yahoo.com (S.W.); ravi22naidu@gmail.com (R.N.); 2School of Medical Sciences, College of Medicine, Nursing and Health Sciences, Fiji National University, Suva, Fiji; vikash.sharma@fnu.ac.fj (V.S.); a.jenney@alfred.org.au (A.J.); 3Fiji Pharmaceutical and Biomedical Services Centre, Ministry of Health and Medical Services, Suva, Fiji; deborahykt@gmail.com; 4Microbiology Unit and Department of Infectious Diseases, The Alfred Hospital, Melbourne, VIC 3004, Australia

**Keywords:** de-escalation, empirical antibiotics, antimicrobial stewardship

## Abstract

**Background:** Antimicrobial resistance is a global health threat and Fiji is not exempt. The appropriate prescribing and timely de-escalation of antibiotics as an integral component of antimicrobial stewardship has been recently introduced in Fiji to help curb antimicrobial resistance through de-escalation, leading to a reduced opportunity for the induction of resistance. **Objectives:** To assess whether empirical antibiotics are being adjusted in a timely fashion in patients admitted with a diagnosis of suspected infection in the Colonial War Memorial Hospital( CMWH) over three months. **Method:** The study was undertaken on patients admitted to the acute medical ward and intensive care unit of the CWMH in Suva (Fiji’s largest hospital). A total of 474 patients were prospectively enrolled at admission when prescribed empiric antibiotic therapy for suspected infections between February and April 2019. **Results:** A total of 356 patients admitted to the Acute Medical Ward and 118 admitted to the Intensive Care Unit were prescribed empiricalantibiotics. These 474 patients were prospectively observed to determine the factors influencing the extent and the timing of antibiotic de-escalation. Only 137 (29%) patients had their antibiotic regimen de-escalated in the first 72 h post-admission based on their microbiological results, whereas, 207 (42%) were de-escalated more than 72 h after admission (OR = 0.5, 95% CI 0.3–0.89; *p* < 0.016). **Conclusions:** At CWMH, antibiotic de-escalation is slow and may be improved by quicker laboratory reporting, greater access to laboratory results for prescribers, and the availability of a wider range of narrow-spectrum antibiotics to assist de-escalation.

## 1. Introduction

Since the introduction of antibiotics, countless lives have undoubtedly been saved through the control of sepsis and infection. However antimicrobial resistance (AMR) has emerged, in great part, as a result of antibiotic misuse and overuse in human healthcare, the agriculture sector, and aquaculture. AMR is a global threat that increases hospital lengths-of-stay, mortality, and substantially, healthcare costs [1,2].

In 2015 AMR was formally recognized as a global health threat of high priority by the 68th World Health Assembly with each member state tasked to develop their own National Action Plan on AMR. The concepts of antimicrobial stewardship (AMS) were introduced to help curb rising AMR rates, and to improve patient safety and outcomes. Empirical antibiotics are prescribed to cover a broad range of suspected infections in medical practice. De-escalation is an integral part of AMS and primarily consists of adjusting initial, broad-spectrum (empirical) treatment by changing the antimicrobial agent to one that is narrower in spectrum or by discontinuing an antimicrobial entirely. These changes are usually guided by the patient’s culture results and clinical progress [3]. Current standards based on international guidelines recommend a review of microbiology cultures and de-escalation of empirical antibiotics by 48–72 h [4].

In 2015, Fiji became one of the first Pacific Island nations to develop a National Action Plan to fight AMR, and this is overseen by the National Antimicrobial Resistance Committee (NARC). It is NARC that is charged with the implementation of AMS activities including awareness engagement, surveillance, research, and infection prevention control, as well as promoting the optimization of the use of antimicrobials, while maintaining national governance in AMS and undertaking sustainable investment action to combat AMR, via coordination of the support of internationally funded programs and regional stakeholders [5] One of the main objectives was to develop an antimicrobial stewardship (AMS) program by 2018 and, as a result, in 2016, an antimicrobial stewardship team was established at CWMH. Fiji has a national antibiotic guideline [6] to set standards and guide prescribers regarding the rationale for the use of antibiotics. Additionally, CWMH has a microbiology laboratory that provides microbiology results that assist in decisions in patient management and antibiotic treatment.

The purpose of this study was to audit current practice and assess whether empirical antibiotics are being adjusted in a timely fashion in patients admitted with a diagnosis of suspected infection to the Acute Medical Ward (AMW) and Intensive Care Unit (ICU) at CWMH over a three-month period.

## 2. Results

During the study period, the AMW and ICU units had 591 total admissions, with 474 of these included in the study with a suspicion of having a bacterial infection necessitating empirical antibiotics. From the AMW, 356 were enrolled (which made up 75% of total AMW admissions) and from the ICU, 118 were enrolled (99% of the total ICU admissions (Figure 1).

Table 1 presents the demographic characteristics of the study population. Among the participants, 51% (240 individuals) were female, and 55% (260) identified as I-Taukei (indigenous Fijian). The predominant age group was 56–75 years, comprising 35% of the study cohort.

Sepsis with an unknown focus was the most common indication for empirical antibiotics (Table 2). 

Table 3 shows that 41% were empirically treated with a third-generation cephalosporin, and ceftriaxone in conjunction with cloxacillin was the most common empirical antibiotic combination prescribed.

A total of 742 microbiological specimens from 432 patients were sent to the laboratory for culture and sensitivity analysis. This included 424 blood cultures, 159 urine cultures, 113 sputum specimens, 35 wound (‘pus’) swabs, 20 stool, and 12 cerebrospinal fluid (CSF) specimens. Bacterial growth was diagnosed in 63 (15%) blood culture sets, 31 (19.5%) urine samples, 48 (42.4%) sputum samples, and 2 (5.7%) wound swabs. There was no growth from any stool or CSF samples collected during this study.

Methicillin-sensitive *Staphylococcus aureus* (MSSA) was identified as the predominant cause of bacteremia (MRSA), comprising 15 cases (3.5%). Fortunately, Methicillin-resistant *Staphylococcus aureus* was found in only 2 (0.5%)samples. Thirty-nine Gram-negative isolates were considered significant in blood culture of which 9 (23.08%) were extended spectrum beta-lactamases (ESBLs) producers (Table 4).

Empirical antibiotic regimens were de-escalated in 137 (29%) and 202 (43%) patients within 72 h and after 72 h, respectively (OR = 0.5, 95% CI 0.3–0.89; *p* < 0.016) (Table 5). We identified 135 patients whose antibiotic regimen was not de-escalated, which included 42 (8.9% of the total enrolled in the study) who had no microbiological specimens taken and 30 (6.3% of total enrolled patients) who died before their culture results were ready, hence were not eligible for de-escalation. Of the remaining 63 patients, 24 (5% of the total enrolled cohort) patients’ microbiology cultures grew a positive microorganism, and 39 (8.2% of the enrolled cohort) patients’ microbiology cultures had no growth.

## 3. Discussion

This prospective observational study showed a suboptimal antibiotic de-escalation rate of 29% within 72 h of admission in patients admitted to the ICU and AMW. Several studies have proven that de-escalation reduces hospital stay and costs, is safe, and is not associated with poor outcomes [7,8,9,10,11,12,13].

The prevalence of de-escalation varies widely in the literature, ranging from 19–68%. It is well documented that the rates are much higher in hospitals in developed countries with well-established antimicrobial stewardship programs [9,14,15,16,17].

The rate of de-escalation observed in this study was similar to that found in a 50-bed ICU in an Indian hospital with a de-escalation rate of 30% [14].

Our findings showed that de-escalation is twice as likely to occur with positive culture results in the >72-h period when compared to the <72-h period. This influence may well be due to culture results being more commonly available or acknowledged later than 72 h after empirical antibiotics were commenced.

Although the reason for this delay was not analyzed in the study, the findings likely reflect later (than ideal) availability, accessibility, and acknowledgment of culture results by clinicians. Another potential reason is that clinicians are not aware of the criteria of de-escalation and do not de-escalate antibiotics at all when a patient’s condition improves [8,9,14].

The rates of culture positivity with *Staphylococcus aureus* are quite high compared to other settings [18,19,20].

This is perhaps something that needs further exploration. The significant number of cultures positive with Gram-negative ESBLs is a concern for our setting as there is limited and sometimes intermittent access to carbapenem antibiotics. The threat of multi-resistant organisms is ever-increasing locally and globally.

It must be taken into account that the study setting is a middle-income country with a very recent introduction of an AMS program at the time of the study. The low rate of de-escalation can be readily improved when good practices are strengthened as the new AMS program becomes established. As part of this strategy, an efficient system to allow ward doctors to view microbiological results early and act on results will definitely improve the timeliness of de-escalation. Additionally, training all clinicians on the AMS program with reinforcement of local antimicrobial guideline use would have an impact.

Educating clinicians on the appropriate use of antibiotics is likely to also impact prescribing practices in terms of when and what to prescribe. A very high proportion of patients (474 of 591 (80.2%)) admitted to the AMW and ICU were given empirical antibiotics. While early de-escalation is good AMS practice, not starting unnecessary antibiotics in the first instance is better. Additionally, there is wide variability in the empirical regimen with some combinations being inappropriate, which is an issue that can be tackled through education and adherence to national guidelines to aid prescribing.

This study was limited by being a relatively short study period of three months; however, with 474 enrolees the observation cohort was large. There was a selection bias in the wards chosen, so the findings of this selective single-center study are difficult to generalize to other departments or centers. Despite this, the study generates potentially useful data on rationalizing antibiotic use in a middle-income country. This study was not able to assess the length of hospital stay and mortality and their association with de-escalation; hence, further, larger prospective studies are recommended to investigate these questions.

## 4. Materials and Method

### 4.1. Study Design 

This was a single-center prospective observational study.

### 4.2. Setting

The study was conducted at the Colonial War Memorial Hospital (CWMH), a tertiary teaching referral hospital for the Central and Eastern Division in Fiji. CMWH has 481 beds with specialist services in medical, surgical, pediatrics, obstetrics, and intensive care.

### 4.3. Participants

From 1 February to 30 April 2019 patients 14 years of age and older, admitted to the AMW and ICU with suspected infection and on empirical antibiotics were enrolled and included in the study. Patients above 14 years who did not receive empirical antibiotics and those in whom infection was not suspected were not enrolled.

### 4.4. Definition of Key Variable

Empirical Antibiotics—antibiotics used to treat an established infection when the causative organism has not been identified.

De-escalation—adjusting initial, broad-spectrum (empirical) treatment by changing the antimicrobial agent to one of a narrower spectrum or discontinuing an antimicrobial combination according to the patient’s culture results.

Timely de-escalation for the purpose of this study was taken as appropriate de-escalation within 72 h of commencement of antibiotics and de-escalation beyond this time frame was considered late.

### 4.5. Data Collection and Analysis

Patients were recruited into the study based on the case notes, using the daily admission lists, and via direct contact with ward registers. These patients were then followed up daily from the time they were admitted to AMW or ICU until they were either discharged or deceased, irrespective of the wards they had later been transferred to, for the duration of inpatient stay.

The practice of medicine at CWMH is team-based and consultant-led, with advanced trainees in medicine (registrars), more junior doctors, medical students, and nursing staff making daily ward rounds to decide upon therapeutic practices such as antibiotic choices The ultimate responsibility for the choice of empirical antibiotics rests with the consultant.

Information regarding the indication for antibiotics, empirical antibiotic regimens, microbiology culture results, and timing to de-escalation was collected by daily inspection of inpatient folders, medication charts, and culture records. Each patient was assessed as to whether the antibiotics prescribed to them could be de-escalated or ceased based on the microbiology culture results.

### 4.6. Statistical Methods

Categorical variables were analyzed through calculation of percentages. The association between the timing of de-escalation (less than 72 h versus greater than 72 h) and positive culture results was assessed using Fisher’s Exact Test. This analysis involved calculating the odds ratio (OR) and 95% confidence interval (CI), along with the corresponding *p*-value, to evaluate the likelihood of timely de-escalation compared to late de-escalation.

Odds ratio was calculated using a statistical calculator found on https://select-statistics.co.uk/calculators/confidence-interval-calculator-odds-ratio (accessed on 27 September 2019).

### 4.7. Ethics

This study was approved by the Ministry of Health and Medical Service Fiji National Health Research and Ethics Committee (FNHRERC 2018.62 CEN) and Fiji National University, College Health and Human Research Ethics Committee. Decision on the type of antibiotics, duration of treatment, and time of de-escalation were made by the treating physician. Data were collected from the patients’ medical folders and there was no direct contact with patients, hence informed consent was waived due to the non-interventional nature of this study.

## 5. Conclusions

This study demonstrates that, in our setting, antimicrobial de-escalation is slower than the standard practice in other places. Therefore, part of the newly introduced AMS program should be aimed at educating all healthcare workers on the importance of antimicrobial de-escalation. Improved timeliness regarding the generation and release of laboratory results and access to a wider selection of narrow-spectrum antimicrobials will give treating teams the tools they need to execute improved de-escalation, and, therefore, better AMS practices.

## Figures and Tables

**Figure 1 antibiotics-14-00124-f001:**
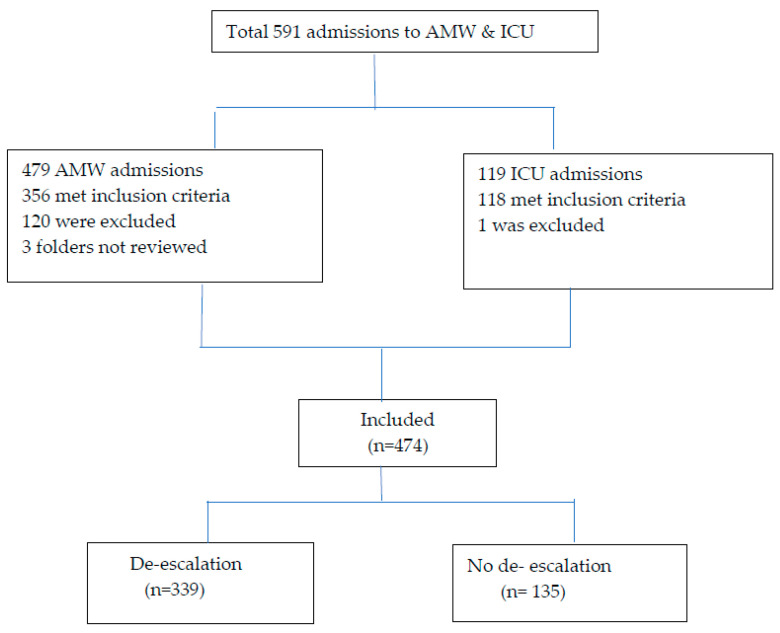
Flow chart of study population inclusion and exclusion process.

**Table 1 antibiotics-14-00124-t001:** Characteristics of patients admitted with suspected infection on empirical antibiotics in AMW and ICU, February to April 2019.

Variable	Total: n = 474 n (%)
Sex	
Male	234 (49)
Female	240 (51)
Age, years	
0–13	0 (0.00)
14–35	113 (23.8)
36–55	161 (33.96)
56–57	165 (34.81)
>76	35 (7.38)
Ethnicity	
I-taukei *	260 (54.85)
Indo-Fijian ^◊^	180 (37.97)
Other	25 (5.27)
Comorbid conditions	
Diabetes	127 (26.79)
Hypertension	101 (21.30)
Autoimmune	17 (3.58)

* I-taukei = Ethnic Fijian, ^◊^ Indo-Fijian = Fijian of Indian descent.

**Table 2 antibiotics-14-00124-t002:** Indication for empirical antibiotics and time to de-escalation.

Indication for Empirical Antibiotics	Total Study Population n (%)	Total cases De-Escalated by 72 h n (%)	Total Cases De-Escalated After 72 h n (%)	No De-Escalation n (%)
Clinical sepsis unknown foci	196 (46%)	58 (30%)	81 (41%)	57 (29%)
Respiratory infection	174 (37%)	52 (30%)	76 (44%)	46 (26%)
Genitourinary infection	12 (3%)	2 (25%)	5 (42%)	4 (33%)
Neurological infection	40 (8%)	11 (28%)	20 (50%)	9 (22%)
Gastrointestinal infection	22 (5%)	8 (36%)	8 (36%)	6 (28%)
Skin and soft tissue	30 (6%)	5 (7%)	12 (40%)	13(43%)
Total	474 (100%)	137 (29%)	202 (43%)	135 (28%)

**Table 3 antibiotics-14-00124-t003:** Most commonly employed antibiotic regimen upon admission.

Antimicrobial Regimen	Clinical Sepsis Unknown Focus n (%)	Respiratory Infection n (%)	Neurological Infection n (%)	Gastrointestinal Infection n (%)	Genitourinary Infection n (%)	Skin and Soft Tissue Infection n (%)
Ceftriaxone + Cloxacillin	103 (56)	42 (24)	19 (48)	6 (17)	6 (50)	4 (13)
Ceftriaxone	26 (13)	7 (4)	4 (10)	0 (0)	0 (0)	0 (0)
Ceftriaxone + Cloxacillin + Acyclovir	0 (0)	0 (0)	7 (18)	0 (0)	0 (0)	
Cloxacillin + Penicillin G + Gentamicin	19 (10)	0(0)	0 (0)	0 (0)	1 (8)	4 (13)
Penicillin G + Cloxacillin	0 (0)	21 (12)	0 (0)	0 (0)	0 (0)	0 (0)
Penicillin G + Doxycycline	0 (0)	34 (20)	0 (0)	0 (0)	0 (0)	0 (0)
Penicillin G + Gentamicin	0 (0)	20 (11)	0 (0)	0 (0)	0 (0)	0 (0)
Ampicillin + Gentamicin + Metronidazole	0 (0)	0 (0)	0 (0)	4 (18)	4 (33)	0 (0)
Penicillin G	0 (0)	16 (9)	0 (0)	0 (0)	0 (0)	0 (0)
Ceftriaxone + Gentamicin	7 (4)	0 (0)	0 (0)	0 (0)	0 (0)	0 (0)
Ceftriaxone + Cloxacillin + Metronidazole	0 (0)	0 (0)	4 (10)	0 (0)	0 (0)	6 (20)
Cloxacillin + Gentamicin + Metronidazole	0 (0)	0 (0)	0 (0)	3 (14)	0 (0)	0 (0)
Cloxacillin + Gentamicin	0 (0)	0 (0)	0 (0)	0 (0)	0 (0)	9 (30)
Others	33 (17)	34 (20)	6 (15)	13 (58)	1 (8)	7 (23)

**Table 4 antibiotics-14-00124-t004:** Microorganism identified in blood culture.

Microorganism	Total Blood Cultures: n = 424 (%)
*Staphylococcus aureus* (MSSA)	15 (3.5)
*Escherichia coli* (non-ESBL)	9 (2.1)
*Klebsiella pneumoniae* (ESBL)	7 (1.6)
*Beta hemolytic Streptococcus*	6 (1.4)
*Salmonella* Typhi	3 (0.7)
*Acinetobacter baumannii*	3 (0.7)
*Klebsiella pneumoniae* (non-ESBL)	4 (0.9)
*Escherichia coli* (ESBL)	2 (0.5)
*Staphylococcus aureus* (MRSA)	2 (0.5)
*Pseudomonas aeruginosa*	2 (0.5)
*Enterobacter cloacae complex*	2 (0.5)
*Serratia marcescens*	1 (0.2)
*Acinetobacter lwoffii*	1 (0.2)
*Citrobacter freundii*	1 (0.2)
*Enterobacter (Klebsiella) aerogenes*	1 (0.2)
*Klebsiella oxytoca*	1 (0.2)
*Proteus mirabilis*	1 (0.2)
*Providencia stuartii*	1 (0.2)
*Streptococcus* spp.	1 (0.2)

**Table 5 antibiotics-14-00124-t005:** 2 × 2 contingency table to calculate the likelihood of positive culture influencing de-escalation within or after 72 h of empiric antibiotic therapy.

	De-Escalation			
	De-Escalation Within 72 h	De-Escalation After 72 h	OR (Odds Ratio) (95% CI)	*p*-Value
Positive Culture	21	54	0.50 (0.30–0.89)	<0.016
Negative Culture	116	148		

## Data Availability

Shared upon reasonable request.

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
