# Peer review of "Prospective Observational Study of De-Escalation of Empirical Antibiotics in Fiji’s National Hospital"

_antibiotics, 2025, doi:10.3390/antibiotics14020124_

Round 1
Reviewer 1 Report
Comments and Suggestions for Authors
The article titled: Prospective observational study of de-escalation of empirical antibiotics in Fiji’s national hospital reports on a nearly 6 year’s old (5 years and 10 month) observation on the rate of de-escalation of antibiotic treatment in patients admitted to the Fiji’s national hospital in February till April 2019. Not considering the fact that the study is outdated, the exact beginning and the end of the study is not indicated (date). The authors did not provide any explanation for selecting this unusual period. They conclude that only 137 (29%) patients had their antimicrobial regimen de-escalated in the first 72 hours post-admission based on their microbiological results whereas, 207 (42%) were de-escalated more than 72 hours after admission.
Regarding the fact that according to Table 3. only 7 different antimicrobials were indicated as initial treatment (plus Metronidazole and Acyclovir) it would have been interesting to know how to de-escalate those regimes and the total length of the antimicrobial treatment. The combination of Penicillin G + Doxycycline disregards the rule that bacteriostatic and bactericide antibiotics should not be combined. Similarly, it does not make any sense to combine a narrow spectrum, beta-lactamase sensitive Penicillin G with a narrow spectrum beta-lactamase resistant penicillin (Cloxacillin), the Cloxacillin should have been appropriate alone. The combination of a third generation cephalosporin (Ceftriaxone) with narrow spectrum penicillin is an unusual treatment regime also.
The dominant bacteria from the blood culture was Staphylococcus epidermidis, which raises the suspicion that this was only a contamination from the skin.
The authors did not make any comparison between the outcomes of the patients (death rate, the length of hospital stay) who were de-escalated after 72 hours or later, or no de-escalation happened.
However the intention of the authors to present the rate of de-escalation of the antibiotic treatment is appropriate, but much more explanation and data would be necessary to accept the article.
Reviewer 2 Report
Comments and Suggestions for Authors
This is a single center retrospective cohort study to evaluate empiric antibiotic use and de-escalation. During three months in 2019, there were 474 patients in their cohort. De-escalation was done in 29% of them within 72 hours and 43% after 72 hours. The rest were not de-escalated. The authors concluded that the de-escalation is slower than standard practice.
This kind of study was done by many people, but this study is valuable to report the de-escalation practice in Fiji, where not many studies were reported from. I would recommend providing more background information related to standard practice in Fiji.
My specific comments are as below.
1. Introduction - I would recommend providing more background information about the practice in Fiji and at CWMH. For example, what has been recommended in Fiji as National Action Plan to fight AMR? Who decides the empiric antibiotic regimen at CWMH? Is there a local treatment guideline for infectious disease? Is there an antimicrobial stewardship team? Is there a microbiology lab at CWMH? And so on. Please add them to the Introduction or Method section.
2. Introduction – it is also recommended to summarize that what kind of antimicrobial stewardship-related study had done in Fiji.
3. Methods - Please add the definition of empirical therapy and de-escalation in the Method section.
4. Methods - What statistical software did they use? Please add the information in the Method section.
5. Table 1 – they categorized patient age 0-15. This group only had 3 patients (as this study included 14 or older). I suggest incorporating this group to next group to create age 14-35.
6. Table 3 – many empirical treatment regimens utilized combination antibiotics but some of them do not make sense to me. (Penicillin G + cloxacillin, Ceftriaxone + cloxacillin etc). Again, is there a local treatment guideline for common infection? In the discussion, the authors stated inappropriate selection of empiric therapy. It would be good to discuss the presence or absence of local treatment guidelines.
7. Results – lines 176-177. This part probably refers to 2x2 table of the timing of de-escalation (within 72 hours and after 72 hours) and culture results (positive and negative) to conduct Fisher’s test and calculate the OR. It is difficult to follow without showing how many patients got positive culture results.
8. Results – Line 177-183. This part is important but is not referred to Table 2.
Round 2
Reviewer 1 Report
Comments and Suggestions for Authors
The corrected version is appropriate and could be accepted in the present form.